# The S100B Protein and Partners in Adipocyte Response to Cold Stress and Adaptive Thermogenesis: Facts, Hypotheses, and Perspectives

**DOI:** 10.3390/biom10060843

**Published:** 2020-05-31

**Authors:** Jacques Baudier, Benoit J Gentil

**Affiliations:** 1Aix-Marseille Université, CNRS, Institut de Biologie du Développement de Marseille, 13009 Marseille, France; 2Departament of Kinesiology and Physical Education and Departament of Neurology and Neurosurgery, McGill University, Montreal, QC H3A 2B4, Canada; benoit.gentil@mcgill.ca

**Keywords:** S100 proteins, p53, ATAD3A, AHNAK, CYP2E1, RPTPσ, enlargosome, exocytosis, annexin2

## Abstract

In mammals, adipose tissue is an active secretory tissue that responds to mild hypothermia and as such is a genuine model to study molecular and cellular adaptive responses to cold-stress. A recent study identified a mammal-specific protein of the endoplasmic reticulum that is strongly induced in the inguinal subcutaneous white adipocyte upon exposure to cold, calsyntenin 3β (CLSTN3β). CLSTN3β regulates sympathetic innervation of thermogenic adipocytes and contributes to adaptive non-shivering thermogenesis. The calcium- and zinc-binding S100B is a downstream effector in the CLSTN3β pathways. We review, here, the literature on the transcriptional regulation of the *S100b* gene in adipocyte cells. We also rationalize the interactions of the S100B protein with its recognized or hypothesized intracellular (p53, ATAD3A, CYP2E1, AHNAK) and extracellular (Receptor for Advanced Glycation End products (RAGE), RPTPσ) target proteins in the context of adipocyte differentiation and adaptive thermogenesis. We highlight a chaperon-associated function for the intracellular S100B and point to functional synergies between the different intracellular S100B target proteins. A model of non-classical S100B secretion involving AHNAK/S100A10/annexin2-dependent exocytosis by the mean of exosomes is also proposed. Implications for related areas of research are noted and suggestions for future research are offered.

## 1. Introduction

Mild-to-moderate hypothermia (28–32 °C) induces cold-shock protein expression and mild endoplasmic reticulum (ER) stress with full activation of the unfolded protein response (UPR) [1,2]. These adaptive responses are sufficient to protect the cells from more severe stress—an effect known as ER hormesis—[3] and are recognized to be neuroprotective [3,4]. At the core of this mechanism of cellular protection is the mitochondria [5]. Deterioration of these homeostatic mechanisms is a general feature of ageing, neurodegenerative disease, and obesity [6,7]. Cellular pathways behind the hormetic response are poorly understood, although increases in the levels of reactive oxygen species (ROS) production by mitochondria and molecular chaperone syntheses are part of mechanisms involved in cold stress protection and recovery [1,4,8,9]. In mammals, adipose tissue is a genuine model to study proteostasis and metabolism associated with cold-stress [10]. Adipose tissues are under the neural control of the sympathetic nervous system, mediated by tyrosine hydroxylase (TH)-positive catecholaminergic neurons that innervate from the paravertebral sympathetic ganglia into adipose tissues [11,12]. Cold exposure stimulates sympathetic nerves to release catecholamine, which then activates adrenergic receptors expressed in adipocytes and stromal cells to trigger lipolysis, white adipose browning, and adaptive thermogenesis [12,13]. Cold exposure also stimulates sympathetic nerve branching, suggesting the existence of positive-feedback regulation [14]. A recent study identified a mammal-specific protein of the ER, called calsyntenin 3β (CLSTN3β) that plays a role in both white (WAT) and brown (BAT) adipose tissues’ sympathetic innervation [15]. CLSTN3β is strongly induced in the inguinal subcutaneous white adipocyte following exposure to cold [15]. CLSTN3β-knockout mice show a defective adaptive thermogenesis, and absence of CLSTN3β expression in adipocytes reduces functional sympathetic innervation in adipose tissues [15]. The zinc (Zn^2+^) and calcium (Ca^2+^) binding protein S100B is a downstream effector of the signaling pathway activated by CLSTN3β in response to the cold [15]. S100B is a member of the S100 family of proteins, the largest group of EF-hand Ca^2+^-binding protein found exclusively in vertebrates [16], and is highly expressed in brain and adipose tissues [17]. In solution, the S100B associates as a non-covalent homodimer with rather low Ca^2+^ affinity (K_D_ 20 µM). S100B also binds Zn^2+^ [18]. Zinc binding studies revealed two sites with strong affinity (K_D_ < 0.1 µM), as well as a variable number of sites with weaker affinities (K_D_ > 1–10 µM). By using coordinating residues on both protomers, zinc ions bridge the dimeric structure of S100B [19], increase the Ca^2+^ affinity of S100B, and promote the Ca^2+^-dependent interactions of the S100B dimer with intracellular target proteins (reviewed in [20]). In addition to two Zn^2+^ equivalents per S100B dimer, weak binding of Zn^2+^ leads to S100B aggregation. Zn^2+^-mediated S100B aggregation contributes to the extracellular function of S100B [21]. The Ca^2+^/Zn^2+^-dependent interaction of S100B with intracellular proteins regulates post-translational modifications such as phosphorylation [22], transcriptional activities [23], enzymatic activities [24], and the assembly state of certain cytoskeletal components and control of their oxidation state [25] (reviewed in [26]). Intracellular S100B also operates within multichaperone scaffolding complexes in adaptive cellular stress responses [20]. Consistent with a direct implication of S100B in adaptive thermogenesis, S100B expression level decreases in warm-acclimatized new-born rat adipose tissues [27], and S100B is up-regulated in progenitor cells committed to thermogenic brown adipocytes [28]. In addition, the *S100b* gene is under the control of the transcription factor PRDM16, which is responsible of the induction of the thermogenic program in brown adipocyte cells [15]. Several recognized or putative intracellular S100B targets (p53, ATAD3A, CYP2E1, AHNAK) harboring consensus S100B binding motifs (Figure 1) have recognized functions in the physiology of adipose tissues. In addition to its intracellular functions, S100B is secreted by adipocytes in response to β-adrenergic receptor stimulation [29,30] where it operates as a neurotrophic factor involved in the sympathetic innervation of thermogenic fat [15]. The paracrine functions of S100B in adipose tissues leaves open two major issues: the mechanism for S100B secretion by adipocytes and the identity of extracellular S100B targets receptors on sympathetic neurons and satellite cells.

Here, we review the links between the transcriptional regulation and interactions of S100B with its intracellular and extracellular targets involved in brown adipocyte differentiation and adaptive thermogenesis. A chaperone-associated function for intracellular S100B in adaptive cold-stress responses and a new model of non-classical S100B secretion by adipocytes by the mean of exosomes are proposed. Finally, we identify two putative receptors targets for extracellular S100B (Receptor for Advanced Glycation End products (RAGE), RPTPσ) harboring consensus S100B binding motifs that may contribute to the extracellular S100B functions in both adipocyte innervation [15] and inflammation associated with obesity [31,32]. Modelling the functions and secretion of the S100B protein in adipocytes should lead to a better understanding of the contributions of brain S100B protein to glial cell differentiation [33], neuron-glia communication [34,35], tissue protection [36], and neural disorders [37].

## 2. Transcriptional Regulation of S100B in BAT Differentiation

S100B is expressed in both WAT and BAT and regulated following a variety of physiological signals [17,27]. S100B expression in adipose tissues is under direct control of the transcription factor PRDM16, a key regulator of BAT differentiation and adaptive thermogenesis [15]. In the immortalized C2C12 mouse myoblast cell line, sustained accumulation of reactive oxygen species (ROS) upregulates S100B [28]. S100B up-regulation cooperates with NF-kB activation to decrease miR-133, a promyogenic and anti-adipogenic factor targeting the degradation of PRDM16 mRNA. As a consequence of the inhibitory effect of S100B on miR-133, PRDM16 is expressed and promotes BAT differentiation [28]. Taken together, these results suggest that PRDM16- and ROS-dependent pathways act synergistically to up-regulate S100B expression in adipocyte in a self-amplification loop (Figure 2). This self-amplification loop likely mobilizes two S100B target proteins, the transcription factor p53 and the oxidative stress-associated Cytochrome P450 2E1 (CYP2E1), that may synergize in order to induce the transcription and translation of PRDM16 and, finally, BAT cell differentiation (see Section 3).

## 3. The Intracellular S100B Targets in BAT Differentiation

Several well-characterized and putative intracellular S100B target proteins (p53, ATAD3A, CYP2E1, AHNAK) have recognized functions in adipocyte differentiation and homeostasis. In Figure 2 and Figure 3, we summarize the interactions of the S100B protein with its intracellular targets in the context of adipocyte differentiation. S100B interaction models take into account ultrastructural studies in epididymal adipocytes showing a cytoplasmic S100B localization in polysomes and ribosomes of the rough ER and accumulation on the outer membrane of mitochondria and in the nuclei of almost all adipocytes [41]. Importantly, S100B does not localize on the Golgi apparatus. The interaction models are also consistent with the hypothesis that S100B operates in multichaperone scaffold complexes to aid in the synthesis and the subcellular sorting of its nuclear (p53) and mitochondrial (ATAD3A and CYP2E1) binding partners [20].

### 3.1. P53

The protein p53 is a short-lived transcriptional regulator that, in response to various forms of cellular stress, is translocated to the nucleus in order to control the expression of a variety of genes involved in metabolism, cell cycle arrest, and cell death [42,43,44]. Nuclear transcriptional functions of p53 requires p53 tetramerization [45]. p53 may also be translocated to the mitochondria where it contributes to mitochondrial metabolism and homeostasis [46] or to specialized contact domains between the ER and mitochondria (mitochondria-associated membranes) where it modulates ER–mitochondria cross-talk [47]. In adipose tissues, p53 is a crucial regulator of adipocytes development, function, and maintenance [48] and exerts a dual activity on WAT and BAT differentiation. While p53 directly drives adipogenic differentiation by increasing the production of mitochondrial reactive oxygen species (ROS) [49], p53 also inhibits WAT differentiation and promotes BAT differentiation [50]. This dual role of p53 in the differentiation of WAT and BAT cells likely reflects the complexity of p53 regulation, stability, subcellular localization (cytoplasmic vs. nuclear), as well as the heterogeneity of its gene targets. Amongst the p53 binding partners, S100B interacts and regulates p53 functions depending of the subcellular localization of the complex (reviewed in [20]). In adipocyte, cytoplasmic S100B may interact with p53 as a co-chaperone to assist in the folding and stability of cytoplasmic p53 monomers prior to p53 nuclear and mitochondria translocation (Figure 2). In addition, S100B can also release p53 of cytoplasmic anchoring protein such as AHNAK (Figure 2). It has been shown that S100B dissociates the AHNAK-p53 complex in the presence of calcium [51] and cooperates with Ca^2+^-dependent protein kinase C (PKC) to promote nuclear p53 translocation and nuclear functions [38,52]. A physical interaction between AHNAK and p53 has also been described in non-small-cell lung cancer cells [39]. In these cells, AHNAK inhibits p53 nuclear function, and ubiquitin-mediated AHNAK degradation is required to activate p53 nuclear activity [39]. Interestingly, in *AHNAK*-/- mice, the number of brown adipocytes increases in white fat tissue [53], which is consistent with the p53 role in BAT adipocyte differentiation. This suggests that, in these mice, AHNAK may not play an inhibitory role on p53 function in pro-BAT differentiation pathways. Studies should further explore the regulation of the p53-AHNAK interaction by Ca^2+^-S100B in BAT cell differentiation [50]. In addition, it is possible that AHNAK may also be involved in S100B secretion in order to promote sympathetic innervation of BAT tissues (see Section 4 and Figure 3).

In the nucleus, p53 tetramers induce the transcription of PRDM16 [50] and the cytochrome p450 2E1 (CYP2E1), a key enzyme involved in the metabolism of nitrosamines and ROS production [54]. Both PRDM16-dependent transcriptional activities and ROS production contribute to p53-dependent adipocyte differentiation [49,50] and might synergize to further increase cellular S100B levels [15,28]. Increase in S100B expression level leads to S100B nuclear accumulation as observed in fully differentiated cells [20,33] and in cancer cells [55]. In the nucleus, S100B inhibits p53 tetramerization [20] and p53 transcriptional activity of the apoptosis program [40], which enhances BAT survival (Figure 2). Hence, concentration-dependent and subcellular localization of S100B in adipocytes may contribute to the p53-dependent adipocyte fate. Further studies are needed to provide more insights into the regulation of S100B protein expression, subcellular localization, and interactions with partners for a better understanding of the S100B–p53 axis in adipocyte fate and development.

### 3.2. ATAD3A

ATAD3A is a nuclear-encoded AAA+-ATPase mitochondrial membrane protein specifically expressed in multicellular eukaryotes (reviewed in [57]). ATAD3A localizes at ER-mitochondria contacts to modulate mitochondria-ER cross-talk [57]. ATAD3A is at the crossroad of processes underlying mitochondrial biogenesis [58] with a direct incidence on the differentiation and fate decisions of progenitors [59,60]. Adipocyte-specific *Atad3a* (Bor) gene deletion in *Drosophila* is responsible of disorganization of the mitochondrial network and of the reduction of cell size, pointing to a key function of ATAD3A in adipocyte cell growth [61]. It has been suggested that S100B operates within multichaperone scaffolding complexes to assist with the de novo synthesis of ATAD3A and its translocation into mitochondria [62]. We propose that, in mammalian BAT cells, S100B assists the de novo ATAD3A protein synthesis to support the need of increased mitochondria biogenesis and thermogenic functions [63] (Figure 3).

By tethering the external and internal mitochondrial membranes and mitochondrial channel components Tom40 and Tim23, ATAD3A is also able to facilitate the transport and degradation of the pro-apoptotic Pink1 protein within the mitochondrial matrix, leading to the down-regulation of parkin-dependent mitophagy, a key E3 ubiquitin–protein ligase [59,60]. Because mitophagy is an important function to maintain appropriate mitochondrial homeostasis in BAT [64], studies on the relationship among S100B levels, ATAD3A processing, and the regulation of Parkin-dependent mitophagy is another molecular pathway that should be considered to provide a better understanding of the contribution of S100B in BAT differentiation.

### 3.3. CYP2E1

CYP2E1 is a key determinant of the cellular redox state and a tightly regulated enzyme [65,66]. In adipose tissues, CYP2E1 is specifically and strongly induced by fasting [67]. *Cyp2e1* gene expression is under the transcriptional control of p53 [54], and as such, likely integrates mitochondrial ROS production associated with p53-dependant adipocyte differentiation [49]. Newly synthesized CYP2E1 localizes at the ER (erCYP2E1) and then is transported to the mitochondria (mtCYP2E1) in a regulated process [68]. Mitochondrial localization of CYP2E1 is responsible for higher levels of ROS and oxidative stress [66].

CYP2E1 is characterized by a consensus S100B-binding motif akin to the one in ATAD3A (Figure 1). Similar to the role of S100B in ATAD3A, further studies should investigate the contribution of S100B in assisting the de novo synthesis of CYP2E1 and mitochondrial addressing of mtCYP2E1 (Figure 2 and Figure 3). In such a scenario, mtCYP2E1 will function in a feedback loop to promote ROS-mediated S100B synthesis [28], and nuclear accumulation will then inhibit p53 transcriptional activity and enhance BAT cell survival (Figure 2).

## 4. S100B secretion by Adipocyte

In addition to intracellular functions, S100B in adipocytes is secreted in response to β-adrenergic receptor stimulation [29,30] and functions as a neurotrophic factor, contributing to sympathetic innervation of thermogenic fat [15]. A recent study has identified calsyntenin 3β (CLSTN3β), a mammal-specific protein of the endoplasmic reticulum, as a key regulator of S100B secretion in brown adipocytes [15]. It has been proposed that CLSTN3β may function as a chaperone for targeting S100B to the ER for subsequent secretion via the trans-Golgi network [15]. A stricto sensus chaperone function for CLSTN3β is challenged by the high ratio of degraded CLSTN3β protein vs. native protein in total cell extracts. More than 50%–80% of the CLSTN3β protein (MW. 39.5 kDa) migrates as a cleaved product or is associated into a high molecular weight complex (see Supplemental Figures 1d, 2c, 3b, and 4a in [15]). Although it is possible that the high molecular weight complexes of CLSTN3β are also interacting partners for S100B, we would rather suggest a function for CLSTN3β in shuttling vesicles containing S100B and/or vesicle exocytosis (Figure 3). We further believe that an understanding of the mechanism of non-canonical S100B secretion must take into account ultrastructural studies of the subcellular localization of S100B in adipocytes, suggesting that S100B secretion is linked to cytoplasmic vesicles rather than the Golgi network [41]. In fact, S100B-positive vesicles fuse with each other or with the plasma membranes to release S100B into the interstitium [41]. A Ca^2+-^dependent exocytosis of S100B by way of secretory vesicles and independent of the ER-Golgi classical secretion pathway has also been described in glioblastoma U87 cells [69]. All these observations suggest that S100B belongs to the leaderless family of secreted proteins trapped within sequestering organelles for interstitial secretion by means of exosomes (reviewed in [70]) (see Figure 3 and Section 4.1). Exosomes can be diverted from their normal exocytotic function and be released as extracellular vesicles (EVs) in pathological conditions [71,72].

### 4.1. Non-Classical Interstitial S100B Secretion by Adipocytes, a Role for AHNAK

In Figure 3, we propose a model of interstitial S100B secretion involving Ca^2+^-dependent exocytosis using the AHNAK1/S100A10/annexin2 complex as a secretory mechanism.

The giant protein AHNAK1 (M.W. 700 kDa) is a downstream effector in β-adrenergic signaling [73,74] and has central functions in adiposity regulations [53,75,76,77]. The AHNAK1 protein has three main functional domains: a short NH2-terminal domain, followed by a large central domain made of repeated amino acid sequences, and a COOH-terminal domain [78]. In AHNAK1, the central repeats interact with S100B [51], whereas the N-terminal and C-terminal domains interact with the S100A10/annexin2 complex [79,80,81]. The S100A10/annexin2/AHNAK1 complex translocates to cholesterol-enriched plasma membranes (rafts) [79], which form a spatial cue for the recruitment and assembly of the components of the exocytotic machinery [82]. When anchored to the plasma membrane, AHNAK1 scaffolds L-type voltage-gated calcium channels (VGCCs), thereby regulating downstream Ca^2+-^dependent pathways necessary for exocytosis [74], and recruits phospholipase C and Ca^2+^-dependent PKC-α required for PI(4,5)P_2_ synthesis [83]. (PI(4,5)P_2_) is a crucial component of the plasma membrane, forming microdomains required for efficient SNARE-mediated exocytosis [84,85]. At the plasma membrane, the AHNAK1/S100A10/Annexin2 complex evolves into vesicles called enlargosomes that are rapidly exocytosed in a SNARE- and Ca^2+^-dependent manner [86,87,88]. During enlargosome biogenesis, AHNAK1 is transported across the enlargosome membrane, apparently by an ABC transporter, and binds to its luminal face [88]. Because S100B protein interacts with AHNAK1’s internal repeats [51], we hypothesize that Ca^2+^-bound S100B could be co-internalized with AHNAK1 within enlargosome-like vesicles and subsequently released into the extracellular spaces (Figure 3). In fact, both S100B and AHNAK1 are constituents of extracellular vesicles (EVs) purified from obese rat adipocyte [89], which confirms co-internalization of S100B and AHNAK1 into exosomes. Interestingly, AHNAK2, which interacts with bFGF, also participates in stress-induced non-classical secretion of bFGF, supporting a role of the AHNAK family in non-classical secretion pathways [90]. Confirmation of the hypothesis on the role of the AHNAK proteins (AHNAK1 and AHNAK2) in the secretion of S100B should be further investigated using genetically modified mouse models.

Finally, it is of prime importance to understand how AHNAK contributes to the exocytotic vesicle (enlargosome) formation. Increasing evidence suggests that the proteolysis of the membrane bound AHNAK protein is key (Figure 3). AHNAK1 is highly sensitive to metal-dependent proteases [91] and calcium-dependent proteolysis leads to the cleavage of AHNAK1 at specific sites on N- and C- terminal domains [92]. AHNAK1 contained in purified EVs migrates as a set of shorter peptides compared to the full length AHNAK1, supporting the hypothesis that AHNAK1 proteolysis accompanies vesicle formation (this evidence is supported by comparing Figure 10 panel c and panel f in [93] and in Figure 7a of [94]). Such a scenario is further supported by the finding that targeting the C-terminal domain of AHNAK1 (residues 5645–5673) to the membrane bound annexin2/S100A10 complex, in place of the full length AHNAK1, was sufficient to cause membrane blebbing that evolves into vesicular structures (Figure 4). In this study, we have overexpressed the minimal S100A10/annexin2 binding domain 1 of AHNAK1 (A2tBP1-residues 5645-5673) in MDCK cells using a plasmid expressing four A2tBP1 sequences in tandem repeats (A2tBP1^4^-EGFP) [80,81]. The intracellular distribution of the A2tBP1^4^-EGFP peptide and annexin2 subcellular localization were studied during Ca^2+^ switch experiments by live fluorescence imaging. As observed with the endogenous full length AHNAK [79], Ca^2+^ addition to the culture medium induces a translocation of A2tBP1^4^-EGFP from the cytoplasm to the plasma membrane within 2 min (Figure 4A). At the plasma membrane, the A2tBP1^4^-EGFP peptide competes with the full length AHNAK for binding to the S100A10/annexin2 complex [80] and induces formation of membrane vesicles (Figure 4B).

We propose that Ca^2+^-dependent AHNAK1 proteolysis disorganizes the AHNAK1/S100A10/Anxa2 scaffolding complex and sub-membrane cytoarchitecture, reducing membrane tension and allowing membrane vesicle formation, with the central domain of AHNAK1 working as a spacer between two lipid rafts [92,95]. Further studies are needed to unravel the apparent complexity of the molecular mechanisms involving CLSTN3β and AHNAK1 in S100B translocation and secretion and more generally to clarify the β-adrenoceptor signaling networks for Ca^2+^-dependent exocytosis of adipokine-containing vesicles [96].

It is important to keep in mind that we cannot exclude the existence of differences in the secretion processes between the WAT and the BAT. Unless the secretion mechanisms are fully characterized, caution should be exercised before generalizing a secretion model for all adipocytes.

### 4.2. Alternative Mechanism for S100B Release from Circulating Extracellular Vesicles

Adipocytes, as well as brain cells, release S100B in circulating fluids by an unknown mechanism [97]. Dynamic S100B release into the blood and into the cerebrospinal fluid during acute brain injury may serve as a repair mechanism [98] but also contribute to neural disorders [37]. EVs are relevant carriers of both S100 proteins [99] and AHNAK [90,93,94,100,101,102]. Hence, an alternative pathway for S100B release into circulating fluids using EVs is plausible. An alternative S100B secretion pathway that mobilizes the Receptor for Advanced Glycation End products (RAGE) has been described [103]. RAGE-mediated S100B-secretion can either occur at the plasma membrane or in the extracellular space [103]. The ectodomain of RAGE harbors a putative S100B-binding NEAL motif [20] onto which extracellular S100B binds [21]. Whether or not S100B plays a role in an autocrine loop to regulate its own secretion needs further investigation.

It is noteworthy that several observations point to a possible interplay between RAGE and AHNAK in adaptive thermogenesis. In fact, both RAGE- and AHNAK-null mice display significant protection from high-fat diet-induced obesity and exhibit a superior ability to thermoregulate during a cold challenge, compared to wild-type mice [53,76,104]. In addition, RAGE stimulation may interfere with β-adrenergic signaling in WAT and BAT cells (106), and β-adrenergic signaling is modulated by AHNAK (74,75). We believe that these data should be taken into account to further investigate the mechanisms controlling S100B secretion and the respective functions of AHNAK and RAGE in adaptive thermogenesis.

## 5. The S100B Interaction with Extracellular Targets

S100B KO mice develop BAT normally, although with impairment of sympathetic innervation (X. Zeng, Harvard Medical School; personal communication). The identity of receptors for the extracellular S100B protein is a key issue in understanding the neurotrophic activity of S100B and the sympathetic innervation of adipose tissue. A strict consensus sequence of S100B-binding motif is present within the fourth fibronectin extracellular domain (FNIII-4, residues 673–694) of the neurotrophic type IIa receptor protein tyrosine phosphatases (PTPRS or RPTPσ) (see Figure 1, and [20]). This domain interacts with proteoglycans and it is anticipated that extracellular soluble factors may regulate the binding of this domain with proteoglycans [105]. In the brain, RPTPσ plays an important role in the regulation of axonal outgrowth and neural regeneration [106,107]. Interestingly, RPTPσ is also highly expressed in adipose tissue [108,109] and represents a strong candidate to mediate the neurotrophic functions of S100B in both the brain and adipose tissues (Figure 3). There is also appreciable in vitro and in vivo evidence that S100B may participate in neuroprotection/neurodegeneration and in brain synaptogenesis via RAGE signaling, although there are uncertainties about the exact mechanisms underlying the contribution of S100B (reviewed in [110] and [37]). The contribution of RAGE to the growth of sympathetic nerve fibers in fat tissue in response to S100B stimulation deserves investigations.

In Figure 3, a paracrine function of S100B on regulatory CD4+ T lymphocytes (T-reg) cells is also suggested. RPTPσ is among 11 unique differentially expressed genes in a sub-population of T-reg cells, which are required for proper BAT activation upon cold exposure [111]. Further studies should investigate the RPTPσ–S100B axis in BAT innervation and in the cross-talk between the immune system and adipose tissue during the development of functional beige fat [112]. A contribution of S100B to paracrine communication between adipocytes and macrophages has also to be considered (Figure 3). In the central nervous system (CNS), S100B is documented to act as an inflammatory cytokine via its interaction with RAGE (For a recent comprehensive review about S100B-activated RAGE signaling in nervous tissue see [37]). Due to the known proinflammatory role of S100B in the CNS it has been proposed that adipose-derived S100B may also play a role in activation of innate immune cells in adipose tissue [97]. In fact, an increase in adipose *S100b* gene expression in WAT is observed during obesity, which, in common with markers of adipose tissue inflammation, can be reversed following weight loss [32]. A specific contribution of the S100B–RAGE axis to paracrine communication in adipose tissue and inflammation has gained further support from in vitro data showing that adipocyte-derived S100B can act as an inflammatory cytokine via RAGE, stimulating M1 polarization of macrophages in an in vitro co-culture system [31]. Further studies are required to understand the complexity of the extracellular function of S100B, AHNAK, RPTPσ, and RAGE in both WAT and BAT and their respective contributions to the cross-talk between adipocytes and immune cells [112].

## 6. Conclusions

Convergent studies suggest a role of S100B in adaptive cellular responses requiring increased cellular metabolism. Beside its role as a chaperone-associated function in adaptive cellular stress responses [20], S100B expression is strongly induced during adaptive thermogenesis following cold stress and may be part of an integrated response to raise energy metabolism (Figure 2). Ca^2+^-dependent interactions of S100B with its binding partners determines its function, and it is interesting to note that three of those partners (ATAD3A, CYP2E1, and p53) are present in adipocytes and play essential roles in mitochondrial metabolism and homeostasis. This suggests that intracellular S100B is a central player that integrates mitochondrial energetics and metabolism together with protein biogenesis in order to promote adipocyte differentiation and thermogenic adaptation. In addition, functional interactions exist between target proteins that can diversify the outcomes of S100B expression. The most striking example of this is the cross-talk between S100B, AHNAK, and p53 (Figure 2 and Figure 3). AHNAK and p53 form a complex in the cytoplasm that can be dissociated in the presence of S100B and calcium (Figure 2). The resulting S100B–p53 or S100B–AHNAK complexes support novel regulatory functions in different cellular compartments (nucleus and secretory vesicles) (Figure 2 and Figure 3). The diversity in the p53–AHNAK–S100B axis functions needs to be studied in more detail and could provide clues to explain the dual activities of p53 and AHNAK on the homeostasis of WAT and BAT cells. In addition to intracellular regulatory functions, S100B has also extracellular neurotrophic activity on sympathetic neurons that contributes to brown adipocyte innervation and adaptive thermogenesis regulation (Figure 3). One candidate to mediate the neurotrophic activity of S100B is the RPTPσ receptor (Figure 3). The contribution of RPTPσ on S100B-dependent sympathetic neurons innervation in adipose tissues and the brain should be further investigated.

In conclusion, the absence of developmental defects in S100B knockout or in double S100B/S100A1 knockout mice have raised questions about the implication of S100B in signaling events related to tissue development. Studies have shown that S100B is not essential for cell viability in physiological situations, but S100B deficiency is harmful under stressful conditions involving high metabolic requirements in the context of cell growth and differentiation during tissue repair [20]. In adipose tissues, S100B is not necessary for cell life and differentiation but is integrated in cellular responses to fine-tune the dynamics of cellular homeostasis by the means of its intracellular and extracellular partners. This supportive function is of particular importance under cold stress-associated adaptive thermogenesis [15].

It is hoped that a better understanding of the functions of S100B in the adipocyte response to cold should also provide information on the molecular basis of the effect of cold shock on the memory disorders observed in transgenic mice overexpressing S100B [113].

## Figures and Tables

**Figure 1 biomolecules-10-00843-f001:**
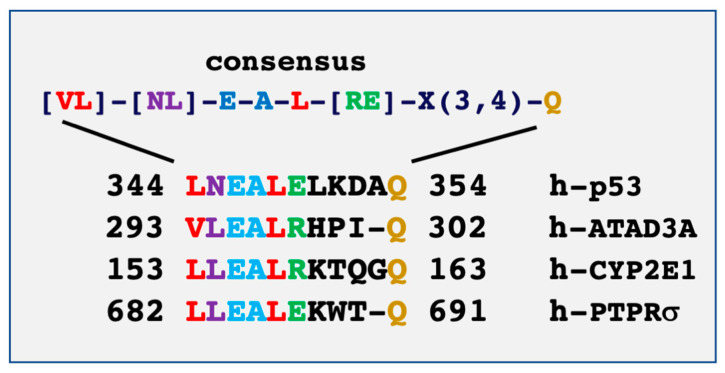
Sequence alignment of the S100B binding domains on p53 and ATAD3A defines a consensus sequence motif called NEAL motif. [20]. The consensus S100B-binding NEAL motif is present in the mitochondrial protein CYP2E1 and in the extracellular receptor protein RPTPσ.

**Figure 2 biomolecules-10-00843-f002:**
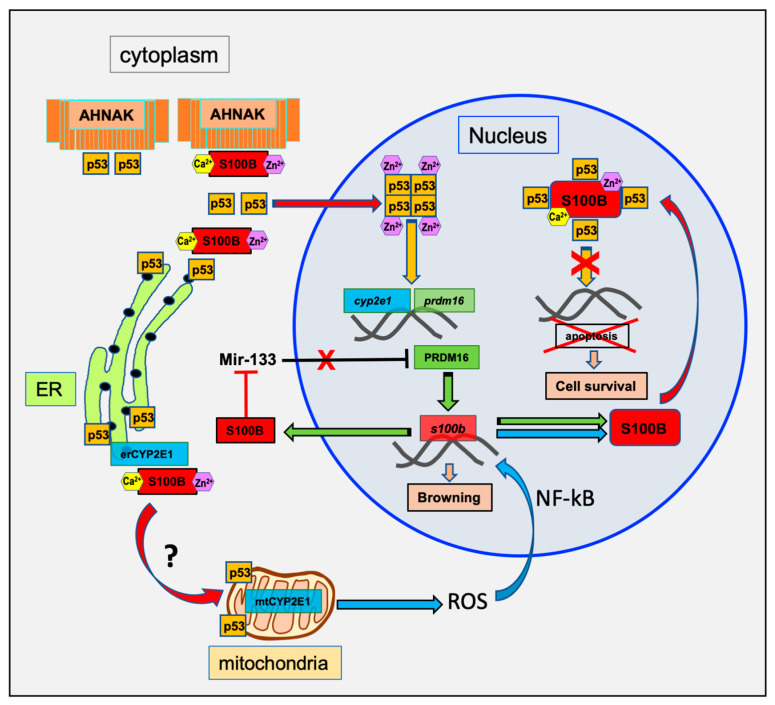
Schematic model of the transcriptional regulation of S100B for brown adipocyte differentiation. Activation of brown adipocyte differentiation triggers S100B transcriptional activation through ROS and PRDM16 pathways [15,28]. Increase in cytoplasmic S100B amplifies nuclear targeting of p53 and p53 transcriptional activation [38] and decreases miR-133 [28]. A negative regulation of p53 through cytoplasmic interaction with AHNAK is also suggested based on a study that characterized physical interactions between AHNAK and p53 and AHNAK-dependent inhibition of p53 nuclear activity [39]. Nuclear translocation of p53 further increases S100B expression level through synergistic transcriptional activation of both PRDM16 and CYP2E1-ROS pathways. Enhanced S100B expression results in S100B nuclear accumulation and down regulation of p53 transcriptional activity [20]. It is hypothesized that the nuclear S100B enhances survival of differentiated BAT by inhibition of nuclear p53-dependant apoptosis [40]. A contribution of S100B to the mitochondria addressing of CYP2E1 and p53 is also suggested (see Section 3 for details).

**Figure 3 biomolecules-10-00843-f003:**
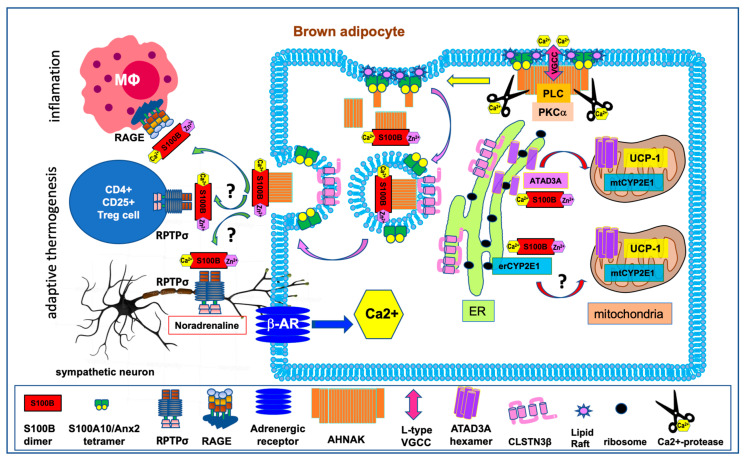
Schematic representation of beta-adrenergic receptors signaling and S100B functions for mitochondrial biogenesis and adipokinesis in brown adipocyte. Mammalian BAT is made up of specialized adipocytes that express uncoupling protein 1 (UCP1), which dissipates the mitochondrial proton gradient, forcing increased flux through the electron transport chain and subsequent heat generation. Brown fat is innervated by nerve endings of the sympathetic nervous system. Sympathetic neurons release noradrenaline molecules that bind to and activate the β3-adrenergic receptors (β-AR) on fat cells. This activation triggers a cascade of biochemical events, such as changes in cytoplasmic Ca^2+^ concentration, elevations of intracellular cAMP levels, activation of kinases cascades, and mitochondrial biogenesis [56]. β-AR stimulation also induces S100B transcription [15]. The cytoplasmic Ca^2+^/Zn^2+^-bound S100B assists the synthesis, folding, stability and/or subcellular addressing of nuclear (p53) or mitochondrial (ATAD3A, CYP2E1) S100B binding partners involved in brown adipocyte differentiation and mitochondrial biogenesis. Changes in cytoplasmic Ca^2+^ concentration also induce AHNAK proteolysis and Ca^2+-^dependent exocytosis of S100B-containing vesicles by means of enlargosomes. In the extracellular space, it is hypothesized that S100B binds to RPTPσ receptors on sympathetic neurons and Treg cells to support the development of functional beige fat. S100B can also bind to RAGE on macrophages. The S100B–RAGE axis contributes to the cross-talk between adipocytes and immune cells and may play a role in inflammation associated with obesity. See also Section 5 for details).

**Figure 4 biomolecules-10-00843-f004:**
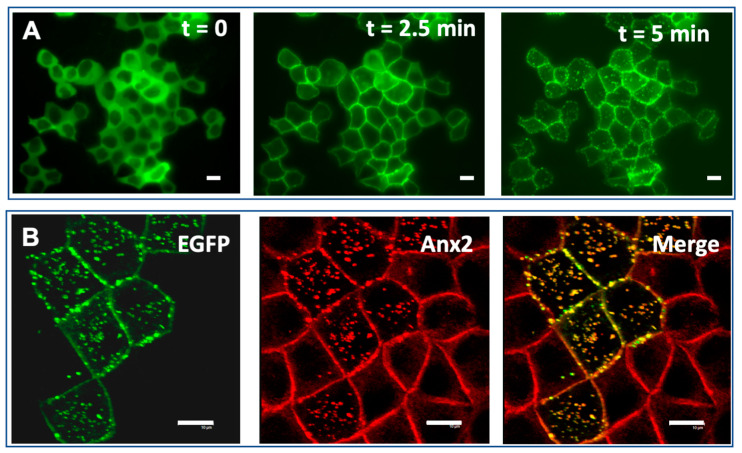
The AHNAK1 C-terminus binding motif specific for the Annexin2/S100A10 (residues 5645-5673) induces formation of membrane vesicles. (**A**) Confluent MDCK cells transfected with plasmid expressing four repeats of the 20-amino acid peptide A2tBP1 (residues 5645-5673) fused to EGFP were incubated in medium containing 5 mM EGTA and supplemented with 1mM MgCl2 for 10 min at 37 °C (t = 0), then shifted to calcium containing medium for 2.5 min (t = 2.5 min) and 5 min (t = 5 min). The subcellular distribution of A2tBP1^4^-EGFP during Ca^2+^ switch experiments was recorded by live fluorescence imaging. (**B**) The translocation of A2tBP1^4^-EGFP to the plasma membrane and vesicles follows that of the annexin2. Confluent MDCK cells transfected with A2tBP1^4^-EGFP (green) were subjected to Ca^2+^ switch experiment, fixed after 10 min and immunostained with annexin2 antibody (red).

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
