# Peer review of "The S100B Protein and Partners in Adipocyte Response to Cold Stress and Adaptive Thermogenesis: Facts, Hypotheses, and Perspectives"

_biomolecules, 2020, doi:10.3390/biom10060843_

Round 1

Reviewer 1 Report

In this interesting review article the authors survey the literature on regulatory effects of S100B in adipocytes and thermogenesis and propose a unitary view of intracellular and extracellular activities of S100B in adipose tissue mostly based on S100B interactions with AHNAK, ATAD3A, p53, and CYP2E1 within adipocytes and with RPTPσ and RAGE outside adipocytes. This review mostly concentrates on the potential role of intracellular S100B as a chaperonin and somewhat neglect S100B’s regulatory effects on e.g. p53 levels (J Biol Chem. 2010 Aug 27;285(35):27487-98), p53 transcriptional activity (J Biol Chem. 2001 Sep 14;276(37):35037-41), MDM2 activity (FEBS Lett. 2010 Aug 4;584(15):3269-74), and NF-kappaB activity (J Cell Physiol. 2010 Apr;223(1):270-82) that might not be ascribed to a chaperonin role of S100B. A review article may certainly convey hypotheses and perspectives on a given biological issue, yet with an eye to additional aspects of the same issue.

Comments

  1. Whereas interactions of S100B with AHNAK, ATAD3A and p53 are well documented, there is no experimental evidence that S100B interacts with CYP2E1 or RPTPσ; only a consensus sequence motif implicated in the S100B binding to AHNAK, ATAD3A and p53 can be found in CYP2E1 and RPTPσ. Thus, the authors should make it clear that any reference to S100B-CYP2E1 and S100B-RPTPσ interactions is but hypothetical (perhaps by putting a question point at any pertinent place in the Figs. of this article).
  2. 2. “Apoptose” should be “Apoptosis”.
  3. The authors represent S100B in a complex with Ca2+ and Zn2+ in their cartoons. While the importance of Ca2+ binding to S100B is universally recognized, how relevant is the interaction of S100B with Zn2+ to the regulatory activities of the protein within and outside cells? The review would benefit from clarification of this point.
  4. Lines 170-185 and Fig. 3. The text and Fig. 3 exclude that S100B can activate RAGE on sympathetic nerve fibers; only S100B-RAGE complex formation in macrophages is being represented. What is the experimental basis for this conclusion? There is abundant literature about RAGE signaling activation by S100B in neurons (and about neurotrophic effects of S100B-activated RAGE signaling in neurons).
  5. Although represented in Fig. 3, RAGE-mediated S100B effects in macrophages (including adipose tissue macrophages) are somewhat neglected in this review. The authors should briefly summarize the available information on S100B-macrophage relationships in adipose tissue.

Author Response

Dear Sir,

Please find a revised version of the manuscript (biomolecules-802075) entitled " The S100B protein and partners in adipocyte response to cold stress and adaptive thermogenesis. Facts, hypotheses and perspectives" submitted to Biomolecules.

The manuscript, which was reviewed by two external reviewers, was considered acceptable in principle for publication in Biomolecules provided that major changes were made to the form and presentation of the original text.

In the revised manuscript, we have tried to respond diligently to all the questions raised by Reviewer 1. Our point by point answers are presented below. Changes to the manuscript are in yellow. We hope you will find the review satisfactory.

Thank you for your help in publishing our review article in Biomolecules.

Best regards.

Reviewer 1.

 Reviewer 1 write: This review mostly concentrates on the potential role of intracellular S100B as a chaperonin and somewhat neglect S100B’s regulatory effects on e.g. p53 levels (J Biol Chem. 2010 Aug 27;285(35):27487-98), p53 transcriptional activity (J Biol Chem. 2001 Sep 14;276(37):35037-41), MDM2 activity (FEBS Lett. 2010 Aug 4;584(15):3269-74), and NF-kappaB activity (J Cell Physiol. 2010 Apr;223(1):270-82) that might not be ascribed to a chaperonin role of S100B. A review article may certainly convey hypotheses and perspectives on a given biological issue, yet with an eye to additional aspects of the same issue.

We agree with the reviewer's main concern. In the revised version, we clearly emphasize in the introduction that S100B interacts with several intracellular proteins for a wide variety of cellular regulatory functions and provide references (P. 2, l. 56-68).

Specific comments.

1. Whereas interactions of S100B with AHNAK, ATAD3A and p53 are well documented, there is no experimental evidence that S100B interacts with CYP2E1 or RPTPσ; only a consensus sequence motif implicated in the S100B binding to AHNAK, ATAD3A and p53 can be found in CYP2E1 and RPTPσ. Thus, the authors should make it clear that any reference to S100B-CYP2E1 and S100B-RPTPσ interactions is but hypothetical (perhaps by putting a question point at any pertinent place in the Figs. of this article).

We have been diligent to make it clear that S100B-CYP2E1 and S100B-RPTPσ interactions is but hypothetical by putting a question point at any pertinent place in Fig. 2 and 3 and in the text when needed (P. 4, l. 124 and P. 6, l. 190-192)

2. “Apoptose” should be “Apoptosis”.

"Apoptose" has been replaced by "Apoptosis" throughout the text.

3. The authors represent S100B in a complex with Ca2+ and Zn2+ in their cartoons. While the importance of Ca2+ binding to S100B is universally recognized, how relevant is the interaction of S100B with Zn2+ to the regulatory activities of the protein within and outside cells? The review would benefit from clarification of this point.

In response to the reviewer's suggestion, we have introduced a short paragraph on the binding properties of S100B to calcium and zinc ions and on the regulation of the interactions of S100B with its intra- and extracellular target proteins. (P. 2, l. 56-68).

4. Lines 170-185 and Fig. 3. The text and Fig. 3 exclude that S100B can activate RAGE on sympathetic nerve fibres; only S100B-RAGE complex formation in macrophages is being represented. What is the experimental basis for this conclusion? There is abundant literature about RAGE signalling activation by S100B in neurons (and about neurotrophic effects of S100B-activated RAGE signalling in neurons).

We agree with the reviewer’s comment. We therefore quote in the revised version of the manuscript a recent review on topic of S100B-activated RAGE signalling in brain and left open the possibility for S100B to activate RAGE on sympathetic nerve fibres in adipose tissues. (P. 10, l. 353-357).

5. Although represented in Fig. 3, RAGE-mediated S100B effects in macrophages (including adipose tissue macrophages) are somewhat neglected in this review. The authors should briefly summarize the available information on S100B-macrophage relationships in adipose tissue.

To comply with the reviewer’s request, we have introduced a short paragraph summarizing the information on S100B-macrophage relationships in adipose tissue. (P. 10, l. 362-375).

Reviewer 2 Report

This is an excellent and informative review demonstrating the links between the transcriptional regulation and interactions of S100B with its intracellular and extracellular targets involved in brown adipocyte differentiation and adaptive thermogenesis. I have no particular comments and recommend to accept the review article for publication in biomolecules.

Author Response

Reviewer 2.

Reviewer 2 accepted the Ms. as it stands.

Round 2

Reviewer 1 Report

The authors have significantly improved the manuscript and satisfied most of my previous criticism. However, a couple of points still remain to be elucidated.

Fig. 2. “Apoptose” still appears in the cartoon (nucleus). “Apoptose” should be corrected to “Apoptosis” therein.

I agree with the authors’ contention that “There are also observations that involve S100B-activated RAGE signalling in the growth of neurites in brain cells, although there are uncertainties about the exact mechanisms underlying the contribution of S100B [110].”. However, there is appreciable in vitro and in vivo evidence that S100B may participate in neuroprotection/neuroregeneration and in synaptogenesis (J Biol Chem 2000; 275: 40096-40105; FASEB J 2004; 18:1812-1817; FASEB J 2004; 18: 1818-1825; J Neurosci Res 2006; 83: 897-906; J Neurochem 2007; 101: 1463-1470; J Cell Sci 2010; 123: 4332-4339; Mol Cell Neurosci 2010; 45: 139-150; J Neurotrauma 2020; 37:1097-1107) via RAGE signaling in part. Above all, it is difficult to imagine that the only raison d'etre of S100B is to promote macrophage/microglia activation to fuel inflammation. In this respect, the authors are right when they state “The contribution of RAGE to the growth of sympathetic nerve fibres in fat tissue in response to S100B stimulation deserves investigations.”.

Author Response

Minor revisions were requested by Reviewer 1, these have been done and changes to the manuscript are in purple.

1- "Apoptose" has been replaced by "Apoptosis" in the revised Figure 2 (p.4).

2- To comply with the reviewer’s request, we have introduced a short sentence that clearly emphasis that S100B may participate in neuroprotection/neuroregeneration and in brain synaptogenesis and we provide adequate references (P. 10, l. 354-358).